# Combination Therapy with Oral Vancomycin Plus Intravenous Metronidazole Is Not Superior to Oral Vancomycin Alone for the Treatment of Severe *Clostridioides difficile* Infection: A KASID Multicenter Study

**DOI:** 10.3390/antibiotics14121252

**Published:** 2025-12-11

**Authors:** Young Wook Cho, Jung Min Moon, Hyeong Han Lee, Jiyoung Kim, Chang Hwan Choi, Kang-Moon Lee, Young-Seok Cho

**Affiliations:** 1Department of Internal Medicine, Dongshin Hospital, Seoul 03653, Republic of Korea; rainbow8454@hanmail.com; 2Department of Internal Medicine, Seoul St. Mary’s Hospital, College of Medicine, The Catholic University of Korea, Seoul 06591, Republic of Korea; lhh2002@hanmail.net; 3Center for Health Promotion and Optimal Aging, Seoul National University Hospital, Seoul 03080, Republic of Korea; 83872@snuh.org; 4Department of Internal Medicine, Chung-Ang University College of Medicine, Seoul 06973, Republic of Korea; gicch@cau.ac.kr; 5Department of Internal Medicine, Changwon Hanmaeum Hospital, Changwon 51139, Republic of Korea; 6Department of Internal Medicine, St. Vincent’s Hospital, College of Medicine, The Catholic University of Korea, Suwon 16247, Republic of Korea; k.jiyoung91@gmail.com

**Keywords:** *Clostridioides difficile*, vancomycin, metronidazole

## Abstract

**Background/Objectives**: Guidelines recommend combination therapy with oral vancomycin and intravenous (IV) metronidazole for fulminant *Clostridioides difficile* infection (CDI). Although patients with severe CDI are often managed with combination therapy, evidence supporting this practice remains limited. This study was performed to compare the clinical outcomes of vancomycin monotherapy versus combination therapy in patients with severe CDI. **Methods**: We conducted a multicenter, retrospective, observational cohort study including adult patients with severe CDI who received oral vancomycin between January 2017 and May 2021. Patients were classified as receiving combination therapy if IV metronidazole was administered for at least 72 h within 48 h of initiating oral vancomycin; otherwise, they were classified as receiving vancomycin monotherapy. The primary outcome was a composite of inpatient all-cause death or colectomy within 60 days after CDI diagnosis. The secondary outcomes were the clinical cure rate, CDI recurrence rate, time to discharge after CDI diagnosis, and duration of ICU admission. **Results**: In total, 215 patients were included, with 100 (46.5%) receiving combination therapy. There were no significant differences in in-hospital mortality or colectomy between the monotherapy and combination therapy groups (25.2% vs. 26.0%, *p* = 1.00). Recurrence rates (19.1% vs. 16.8%, *p* = 0.81), total length of stay (31.0 vs. 23.0 days, *p* = 0.16), and ICU stay duration (35.0 vs. 32.0 days, *p* = 0.89) were also similar. However, the clinical cure rate was significantly higher in the monotherapy group than in the combination therapy group (79.1% vs. 65.0%, *p* = 0.03). **Conclusions**: Combination therapy with oral vancomycin and IV metronidazole was not associated with improved clinical outcomes in patients with severe CDI. Prospective randomized studies are needed to clarify optimal management strategies for severe CDI.

## 1. Introduction

*Clostridioides difficile* infection (CDI) is a common condition worldwide and a leading cause of healthcare-associated infectious diarrhea in North America. CDI is associated with increased morbidity and mortality [1]. A recent systematic review reported that the incidence of CDI among hospitalized patients in Asia ranges from 2.60 to 25.20 cases per 10,000 patient-days, suggesting that CDI is also a significant nosocomial pathogen in these regions [2]. A recent study using nationwide claims data in the Republic of Korea showed that the incidence of CDI has gradually increased from 0.3 per 1000 admissions in 2008 to 1.6 in 2020, indicating that recent rates are approaching those in Western countries [3]. In addition, the CDI-attributable length of hospital stay and medical costs have increased considerably in Republic of Korea [4].

Several clinical practice guidelines recommend managing patients with CDI according to disease severity. For patients with severe complicated or fulminant CDI, the Infectious Diseases Society of America and the Society for Healthcare Epidemiology of America (IDSA/SHEA) guidelines as well as the American College of Gastroenterology guidelines recommend adding intravenous (IV) metronidazole and, in cases of paralytic ileus, vancomycin enema to oral or rectal vancomycin [5,6]. These recommendations are based on expert opinion and a single-center, retrospective, observational comparative study involving 88 critically ill patients with CDI in the intensive care unit (ICU), which showed lower mortality in patients receiving combination therapy than in those receiving vancomycin monotherapy [7]. However, subsequent studies have not demonstrated improved clinical outcomes with combination therapy compared to vancomycin alone [8,9]. Based on these findings, the 2021 European Society of Clinical Microbiology and Infectious Diseases (ESCMID) guidelines do not recommend the routine addition of IV metronidazole to oral antibiotic therapy for severe CDI and instead recommend either oral vancomycin or fidaxomicin monotherapy [10]. To date, because of ethical considerations, randomized controlled trials have not been available to support the benefit of combination therapy in treating severe or fulminant CDI. The aim of this study was to evaluate the clinical outcomes of vancomycin monotherapy versus combination therapy with vancomycin and IV metronidazole in patients with severe CDI.

## 2. Results

### 2.1. Patient Characteristics

In total, 215 patients were included in the study: 115 patients (53.5%) received oral vancomycin alone, and 100 (46.5%) received combination therapy. The study flow is shown in Figure 1. Patients with missing data (*n* = 9) or who received CDI treatment more than 2 days before CDI diagnosis (*n* = 2) were excluded. Table 1 summarizes the characteristics of the study population. Their mean age was 72.0 ± 14.5 years. Women were significantly more represented than men in the combination therapy group (60.0% vs. 40.0%, *p* = 0.016). Among all included patients, toxin EIA was performed in 91 (42.3%). Overall, the toxin positivity rate was 64.8%, with no significant difference between the two groups (64.0% vs. 65.9%, *p* = 0.91). Other demographic characteristics did not differ significantly between the groups.

### 2.2. Outcomes

Table 2 summarizes the treatment outcome measures. Overall, 26% of patients met the primary outcome of death or colectomy within 60 days. The rates of in-hospital mortality or colectomy were not significantly different between the two groups (25.2% vs. 26.0%, *p* = 1.00). Twenty-seven patients (23.5%) in the monotherapy group and 25 patients (25.0%) in the combination group died (*p* = 0.92). Two patients (1.7%) in the monotherapy group and one patient (1.0%) in the combination group underwent colectomy (*p* = 1.00). The clinical cure rate was significantly higher in the monotherapy group than in the combination therapy group (79.1% vs. 65.0%, *p* = 0.03). CDI recurrence rates were similar regardless of treatment regimen (19.1% vs. 16.8%, *p* = 0.81). There were no significant differences in the total length of stay (31.0 vs. 23.0 days, *p* = 0.16) or intensive care unit (ICU) length of stay (35.0 vs. 32.0 days, *p* = 0.99) after CDI diagnosis between the two groups. There were also no differences in primary or secondary outcomes according to toxin positivity (Table 3). Multivariable analysis identified four factors independently associated with 60-day death or colectomy: heart rate, blood urea nitrogen level, albumin level, and hematologic malignancy (Table 4).

## 3. Discussion

The aim of the present study was to evaluate the efficacy of combination therapy with oral vancomycin and IV metronidazole on mortality or colectomy in patients with severe CDI. We found that the composite CDI-related outcomes of death or colectomy, as well as recurrence, did not differ between the two groups. Our findings are consistent with those of previous studies [8,9,11]. Our results showed significant higher clinical cure rate at 10 days compared with those of previous studies, suggesting that vancomycin monotherapy would be better for the treatment of patients with severe CDI. This discrepancy could be attributed to a difference in the study population. However, better treatment response should be confirmed in further well-designed prospective study. In addition, four factors, including increased heart rate, high blood urea nitrogen level, low serum albumin level, and hematologic malignancy were independently associated with 60-day death or colectomy in multivariable analyses. These findings were similar with results from a previous study that the variables most strongly associated with death or colectomy included increased baseline comorbidities (Charlson comorbidity index > 7 points), hospitalization in the ICU at the time of index *C. difficile* test, and low serum albumin level [9].

Many clinicians typically manage patients with severe complicated or fulminant CDI using a combination of oral vancomycin and IV metronidazole [12,13]. In 1994, Olson et al. [14] reported results from a 10-year CDI surveillance and treatment study (1982–1992) conducted at the Minneapolis VA Medical Center. In that study, seven of eight patients who had CDI with severe ileus received IV metronidazole in addition to standard oral treatment, and six showed a clinical response. Since then, metronidazole has been widely used in the management of CDI. A retrospective observational study of 88 critically ill patients by Rokas et al. [7] showed that patients in the combination therapy group had significantly lower mortality rates than those receiving monotherapy (36% vs. 16%, *p* = 0.03). Based on findings from such studies, most medical society guidelines recommend adding IV metronidazole (500 mg every 8 h) in conjunction with vancomycin [15]. A recent systematic review of CDI treatment guidelines reported that eight guidelines recommended IV metronidazole in addition to oral vancomycin, with a concordance rate of 80–100% for first-episode fulminant CDI [16]. Similarly, guidelines in Taiwan recommend a combination of oral vancomycin (125–500 mg) and IV metronidazole (500 mg every 8 h) for fulminant CDI [17]. Japanese guidelines recommend combination therapy with IV metronidazole for patients not responding to vancomycin [18]. An international cross-sectional survey of 395 clinicians from 31 countries across six continents found that oral vancomycin was the most commonly used agent (92.9%), often combined with IV metronidazole (75.5%) for fulminant CDI without ileus [13].

However, more recent studies have not shown a statistically significant difference in mortality between combination therapy and oral vancomycin monotherapy. A two-center, retrospective observational study of 2114 patients with non-severe (*n* = 727), severe (*n* = 861), and fulminant CDI (*n* = 526) found that combination therapy for either non-fulminant or fulminant infections was not superior to vancomycin monotherapy and offered no advantage in clinical outcomes such as 90-day mortality, colectomy, or CDI recurrence [9]. Another retrospective study of 138 ICU-admitted patients with non-fulminant CDI showed that overall inpatient mortality was higher in the combination therapy group, while 30-day mortality did not differ significantly between the two regimens [8]. Bass et al. [11] likewise demonstrated no difference in clinical cure rates between monotherapy and combination therapy in patients with severe CDI. A meta-analysis including 190 patients from four studies also reported no significant difference in clinical cure rates between monotherapy and combination therapy [19]. Similarly, a recent meta-analysis of three studies comparing vancomycin alone with combination treatment for fulminant CDI showed only a small, non-significant (2.7%, *p* = 0.8) difference in mortality between the two groups [20]. Taken together, these findings support the 2021 ESCMID guidelines, which do not recommend adding IV metronidazole to standard oral treatment for severe complicated or fulminant CDI [10]. Furthermore, combination therapy does not appear to provide additional benefit in immunocompromised patients. Korayem et al. [21] found no difference in recurrence rates among patients with CDI who had undergone solid organ transplantation and were treated with either vancomycin alone or vancomycin plus metronidazole. A retrospective cohort study evaluating CDI outcomes in hospitalized patients with hematologic malignancies likewise found no difference in clinical resolution among the three treatment modalities—metronidazole, vancomycin, and combination therapy [22]. In addition, a prospective cohort study comparing oral metronidazole, IV metronidazole, and oral vancomycin for mild CDI reported higher mortality in the IV metronidazole group than in the other two groups, suggesting that patients receiving IV metronidazole may experience lower drug exposure [23].

In patients with severe ileus or toxic megacolon due to fulminant CDI, the efficacy of oral vancomycin may be reduced because impaired gastrointestinal motility can prevent or delay the drug from reaching the colon. In such cases, the addition of IV metronidazole may be helpful because intravenously administered metronidazole can achieve therapeutic concentrations in the inflamed colon [6]. Vancomycin may also be delivered via rectal administration as an adjunct to oral therapy, although the quality of evidence supporting this approach is very low [17]. An in vitro study evaluating the antimicrobial interaction between metronidazole and vancomycin against *C. difficile* found no evidence of synergistic effects between the two agents [24]. Similarly, an in vivo mouse study reported that combination therapy with metronidazole and vancomycin did not improve clinical outcomes compared with vancomycin monotherapy [25]. Another study also found that the combination of vancomycin and metronidazole did not affect the survival rate of mice with primary CDI compared with either drug alone [26].

In our study, CDI was diagnosed using two different methods: toxin EIA or NAAT. Although the impact of toxin positivity on clinical outcomes remains controversial, several studies have reported that patients with positive NAAT but negative toxin EIA results experience lower mortality and fewer CDI-associated complications than those in whom both tests are positive [27]. However, a more recent meta-analysis found that patients who tested positive only by NAAT had a risk of CDI-related complications and mortality comparable to that in patients with both positive results [28]. Our findings also showed no significant differences in mortality or complications according to toxin positivity. The primary diagnostic method for CDI has been shifting from toxin EIA to NAAT in several United States hospitals. Although NAAT is highly sensitive for detecting *C. difficile* toxin genes, it may also identify cases of colonization rather than active infection, potentially leading to an apparent increase in the incidence of CDI [6]. In Republic of Korea, NAAT-based diagnosis has been widely adopted since its introduction in 2015, particularly in tertiary and general hospitals. However, a recent epidemiologic study did not observe an abnormally high increase in CDI incidence following its introduction [3].

Another issue to consider is that of which IV agent—metronidazole or tigecycline—should be combined with standard oral antibiotics, although the decision to use a combination regimen remains case-dependent [29]. Tigecycline, a broad-spectrum tetracycline derivative, has demonstrated in vitro activity against *C. difficile* [30]. A retrospective observational cohort study reported that patients receiving IV tigecycline monotherapy (*n* = 45) had significantly better clinical cure rates (75.6% vs. 53.3%, *p* = 0.02), a less complicated disease course (28.9% vs. 53.3%, *p* = 0.02), and less severe sepsis (15.6% vs. 40.0%, *p* = 0.09) than patients receiving vancomycin plus IV metronidazole (*n* = 45) [31]. Several retrospective cohort studies also suggest that tigecycline could be considered a potential therapeutic option for patients with severe CDI [32]. However, the updated ESCMID guidelines only weakly recommend the addition of IV tigecycline to standard oral antibiotics on a case-by-case basis, given the absence of randomized clinical trials [10].

To our knowledge, this is the first study to compare oral vancomycin monotherapy with combination therapy of oral vancomycin and IV metronidazole for the treatment of severe CDI in the Asia-Pacific region. Our study population included patients from three distinct high-volume centers, and CDI diagnosis was not based solely on NAAT. However, several limitations should be acknowledged. First, this was a retrospective review of EMRs and, as such, it is subject to inherent bias. A thorough analysis could not always be performed because of incomplete records. Data regarding CDI severity classification and prior antimicrobial use were obtained through EMR review. Second, the effects of various treatment regimens for severe CDI on clinical outcomes were not fully assessed because few patients received higher doses of oral vancomycin, rectal vancomycin, or consistent treatment durations. Third, relatively high percentage of patients with severe CDI in our study had combination therapy. However, percentage of combination therapy was not much different between our study and previous studies comparing the effect of vancomycin monotherapy and combination therapy with IV metronidazole [8,9,11]. Fourth, patients with fulminant CDI were excluded from our study. Further research is needed to evaluate the efficacy of combination therapy in fulminant CDI, particularly in patients with ileus, for whom combination therapy may be one of the limited available options. Fifth, our study did not account for *C. difficile* strain types, which can influence clinical outcomes. The low prevalence of hypervirulent strains in the Republic of Korea may have affected our findings [33]. Sixth, some potential risk factors may have been overlooked in the statistical analysis because of the relatively small sample size. Finally, because our patients were hospitalized in tertiary referral centers, the cohort likely included individuals with more comorbidities and complications, as well as a high proportion of immunocompromised patients, limiting the generalizability of our results.

## 4. Materials and Methods

### 4.1. Patient Characteristics

We conducted a multicenter, retrospective cohort study at three teaching hospitals in the Republic of Korea: Seoul St. Mary’s Hospital, St. Vincent’s Hospital, and Chung-Ang University Hospital. Data were collected from electronic medical records (EMRs) for adult patients (≥18 years old) who developed CDI between January 2017 and May 2021. A CDI diagnosis was made based on either a positive toxin enzyme immunoassay (EIA) or nucleic acid amplification test (NAAT) result in patients who had been hospitalized for another condition for at least 2 days before symptom onset (defined as at least three unformed stools). The severity of CDI was determined according to the 2017 IDSA/SHEA guidelines, using leukocyte counts and renal criteria [34]. Severe CDI was defined as a white blood cell count of ≥15 × 10^3^/µL or a creatinine level ≥ 1.5 times the baseline. Fulminant CDI, previously referred to as severe complicated CDI, was defined by the presence of at least one of the following: hypotension or shock, ileus, or megacolon. The study included patients with severe CDI who were treated with either oral vancomycin monotherapy or combination therapy with oral vancomycin and IV metronidazole for at least 72 h, initiated within 48 h of starting oral vancomycin. Patients were excluded if they were under 18 years old, had mild-to-moderate or fulminant CDI, received CDI treatment more than 2 days before diagnosis, were treated with oral metronidazole, or had missing data.

### 4.2. Data Collection

Patient information was obtained from EMRs. Collected data included demographics, comorbidities, previous hospitalizations, antibiotic treatments and surgeries, dates and results of CDI diagnostic tests, drugs used (including doses and administration routes), treatment duration, concomitant medications (such as antibiotics, immunosuppressants, or chemotherapy), ICU admission, and length of hospital stay. Clinical variables included vital signs (body temperature, heart rate, respiratory rate, and blood pressure), complete blood counts, biochemical parameters such as serum albumin and creatinine levels, liver function test results, and findings from sigmoidoscopy or colonoscopy, if performed. An immunocompromised state was defined by any of the following: neutropenia (<1000 white blood cells/mm^3^); prior allogeneic stem cell or solid organ transplant; chemotherapy; treatment with an immunosuppressant such as azathioprine, cyclosporine, TNF-alpha inhibitors, mycophenolate mofetil, tacrolimus, or other specific monoclonal antibodies; or prolonged corticosteroid use at a minimum dose of 0.5 mg/kg/day for at least 4 weeks (prednisone or equivalent) within the past 3 months.

### 4.3. Outcomes

Patients were followed for 60 days after CDI diagnosis. The primary outcome was a composite of in-hospital all-cause death or colectomy within 60 days of CDI diagnosis. Composite outcome included colectomy since patients with CDI who underwent colectomy had mortality rate of 50% over [35]. Colectomy events were identified through EMRs within this period. Secondary outcomes included the clinical cure rate at day 10 of treatment (defined as having less than 3 unformed bowel movements in 24 h within 10 days of treatment and sustained response until at least 10 days of treatment), CDI recurrence rate within 60 days of treatment, time to discharge after CDI diagnosis, and duration of ICU admission. Clinical cure was defined as the resolution of diarrhea for at least 48 h within 10 days of treatment. Resolution of diarrhea was defined as fewer than three unformed bowel movements in 24 h within the 10-day treatment period. Recurrence was defined as the re-emergence of diarrheal symptoms requiring retreatment for a new CDI diagnosis within 60 days after the initial diagnosis. If patients were discharged before the end of follow-up, they or their relatives were contacted by telephone or at outpatient clinics to confirm symptoms and vital status within 60 days (at home, hospitalized, or deceased).

### 4.4. Statistical Analysis

For comparisons between the two groups, categorical variables were evaluated using the chi-square (χ^2^) test or Fisher’s exact test, as appropriate. Continuous variables were analyzed using Student’s *t*-test or the Mann–Whitney U test. Univariate Cox regression and stepwise multivariate Cox regression analysis were used to predicting significant independent factors for 60-day dearth or colectomy. All tests were two-tailed, and *p* values of <0.05 were considered statistically significant. All analyses were performed using the Statistical Package for the Social Sciences (SPSS), version 24.0 (IBM, Armonk, NY, USA).

## 5. Conclusions

Our data suggest that combination therapy with oral vancomycin and IV metronidazole is not associated with improved clinical outcomes in severe CDI in Republic of Korea. However, prospective randomized studies are needed to clarify the optimal management of patients with severe CDI.

## Figures and Tables

**Figure 1 antibiotics-14-01252-f001:**
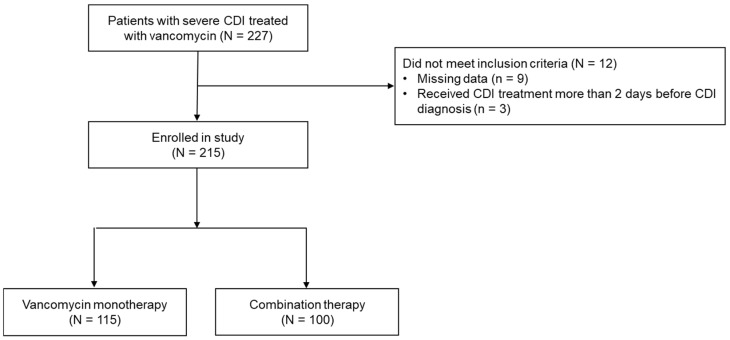
Study flow diagram.

**Table 1 antibiotics-14-01252-t001:** Demographic characteristics.

Baseline Characteristics	Total (*n* = 215)	Monotherapy (*n* = 115)	Combination Therapy (*n* = 100)	*p* Value
	N (%)	N (%)	N (%)	
Sex				
Male	106 (49.3)	66 (57.4)	40 (40.0)	0.016
Female	109 (50.7)	49 (42.6)	60 (60.0)	
Age, y	72.0 ± 14.56	72.2 ± 14.5	71.7 ± 14.5	0.812
Hospital characteristics at the time of CDI diagnosis				
Vital signs (Mean ± SD)				
Systolic BP	122.2 ± 19.9	124.2 ± 21.4	120.0 ± 17.9	0.112
Diastolic BP	68.3 ± 12.0	69.6 ± 11.8	66.8 ± 12.2	0.093
Temperature	37.1 ± 0.7	37.1 ± 0.7	37.2 ± 0.7	0.682
Heart rate	89.6 ± 19.0	88.4 ± 19.8	90.9 ± 18.0	0.333
Laboratory results (Mean ± SD)				
White blood cell count, 10^9^/L	17.2 ± 11.0	16.3 ± 11.5	18.4 ± 10.3	0.157
Hemoglobin, g/dL	9.6 ± 1.6	9.5 ± 1.7	9.8 ± 1.5	0.294
Platelet count, 10^9^/L	106.0 ± 119.6	10.4.2 ± 115.1	108.1 ± 125.2	0.811
BUN, mg/dL	41.5 ± 28.1	41.3 ± 23.6	41.7 ± 32.6	0.921
Creatinine, mg/dL	2.9 ± 2.5	3.0 ± 2.5	2.8 ± 2.5	0.485
Albumin, g/dL	2.6 ± 0.5	2.7 ± 0.6	2.6 ± 0.5	0.440
Toxin EIA				
Negative	18 (15.7)	14 (14.0)		0.919
Positive	32 (27.8)	27 (27.0)		
Comorbidities				
Hematologic malignancies	25 (11.6)	14 (12.2)	11 (11.0)	0.956
Solid cancers	19 (8.8)	8 (7.0)	11 (11.0)	0.423
Solid organ transplantations	11 (5.1)	6 (5.2)	5 (5.0)	1.000
Hypertension	108 (50.2)	61 (53.0)	47 (47.0)	0.455
Coronary artery disease	18 (8.4)	13 (11.3)	5 (5.0)	0.156
Diabetes mellitus	80 (37.2)	41 (35.7)	39 (39.0)	0.715
Liver cirrhosis	8 (3.7)	6 (5.2)	2 (2.0)	0.290
Chronic renal disorders	99 (46.0)	60 (50.2)	39 (39.0)	0.073
Cerebrovascular diseases	45 (20.9)	25 (21.7)	20 (20.0)	0.885
Immunosuppressed				
Immunosuppressants	15 (7.0)	8 (7.0)	7 (7.0)	1.000
Corticosteroids	37 (17.2)	21 (18.3)	16 (16.0)	0.797
ICU admission	60 (27.9)	29 (25.2)	31 (31.0)	0.940
Antibiotic exposures	191 (93.6)	104 (93.7)	87 (93.5)	1.000

BP, blood pressure; BUN, blood urea nitrogen; EIA, enzyme immunoassay; ICU, intensive care unit; SD, standard deviation.

**Table 2 antibiotics-14-01252-t002:** Treatment Outcomes.

60-Day Outcome	Total (*n* = 215)	Monotherapy (*n* = 115)	Combination Therapy (*n* = 100)	*p* Value
	N (%)	N (%)	N (%)	
Death or colectomy	55 (25.6)	29 (25.2)	26 (26.0)	1.000
Death	52 (24.2)	27 (23.5)	25 (25.0)	0.920
Colectomy	3 (1.4)	2 (1.7)	1 (1.0)	1.000
Clinical cure at Day 10	156 (72.6)	91.1 (79.1)	65 (65.0)	0.031
Recurrence	37 (18.0)	21 (19.1)	16 (16.8)	0.814
Length of stay after CDI diagnosis, days, mean (range)	26.0 (2–779)	31.0 (2–779)	23.0 (3–619)	0.159
Length of ICU stay after CDI diagnosis, days, mean (range)	35.0 (3–619)	35.0 (6–112)	32.0 (3–619)	0.988

CDI, *Clostridioides difficile* infection; ICU, intensive care unit.

**Table 3 antibiotics-14-01252-t003:** Treatment outcomes according to toxin EIA positivity.

60-Day Outcome	Total (*n* = 91)	Negative (*n* = 32)	Positive (*n* = 59)	*p* Value
	N (%)	N (%)	N (%)	
Death or colectomy	19 (20.9)	8 (25.0)	11 (18.6)	0.658
Clinical cure at Day 10	65 (71.4)	20 (62.5)	45 (76.3)	0.252
Recurrence	21 (23.1)	7 (21.9)	14 (23.7)	1.000
Length of stay after CDI diagnosis, days, mean (range)	43.5 (3–358)	35.2 (5–144)	47.7 (3–358)	0.352
Length of ICU stay after CDI diagnosis, days, mean (range)	37.2 (4–141)	24.3 (5–86)	45.1 (4–141)	0.081

CDI, *Clostridioides difficile* infection; ICU, intensive care unit.

**Table 4 antibiotics-14-01252-t004:** Factors associated with 60-day dearth or colectomy.

Variable	Univariate		Multivariable Model	
	OR (95% CI)	*p* Value	OR (95% CI)	*p* Value
Treatment				
Combination	Reference			
Vancomycin	1.04 (0.56, 1.93)	0.896		
Sex (female vs. male)	1.12 (0.6, 2.06)	0.727		
Age, y				
18–55	Reference			
56–75	1.39 (0.13, 14.78)	0.785		
≥76	1.33 (0.14, 12.37)	0.804		
Vital signs				
Systolic BP	0.99 (0.97, 1)	0.097		
Diastolic BP	0.97 (0.95, 1)	0.057		
Temperature	0.91 (0.58, 1.43)	0.68		
Heart rate	1.04 (1.02, 1.05)	<0.001	1.03 (1.01, 1.05)	<0.001
Laboratory results				
White blood cell count	1 (1, 1)	0.073		
Hemoglobin	1 (1, 1)	0.067		
Platelet count	1 (1, 1)	0.373		
BUN	1.02 (1, 1.03)	0.005	1.01 (1, 1.03)	0.019
Creatinine	1.02 (0.91, 1.16)	0.699		
Albumin	0.35 (0.18, 0.66)	0.001	0.34 (0.17, 0.7)	0.003
Toxin EIA positivity	0.69 (0.24,1.93)	0.478		
Hematologic malignancies	3.82 (1.62, 8.99)	0.002	3.6 (1.36, 9.51)	0.01
Immunosuppressed				
Immunosuppressants	1.06 (0.32, 3.48)	0.92		
Corticosteroids	1.75 (0.82, 3.74)	0.147		
Antibiotic exposures	4.37 (0.55, 34.47)	0.162		

## Data Availability

Data will be available upon request from the corresponding author.

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
