# Peer review of "Combination Therapy with Oral Vancomycin Plus Intravenous Metronidazole Is Not Superior to Oral Vancomycin Alone for the Treatment of Severe Clostridioides difficile Infection: A KASID Multicenter Study"

_antibiotics, 2025, doi:10.3390/antibiotics14121252_

Round 1

Reviewer 1 Report

Comments and Suggestions for Authors
  1. This study demonstrates a better efficacy of vancomycin monotherapy than metronidazole (Table 2 and rows 123-124). Previous studies cited by the authors show that the two therapeutic regimens have the same efficacy. In my opinion, a brief discussion of the statistically demonstrated superiority of vancomycin monotherapy would be useful.
  2. In my opinion, the four independent factors identified as being associated with 60-day mortality and colectomy (heart rate, blood urea nitrogen level, albumin level, and hematologic malignancy) should also be mentioned in the discussion chapter.

Author Response

Comments 1: This study demonstrates a better efficacy of vancomycin monotherapy than metronidazole (Table 2 and rows 123-124). Previous studies cited by the authors show that the two therapeutic regimens have the same efficacy. In my opinion, a brief discussion of the statistically demonstrated superiority of vancomycin monotherapy would be useful.

Response 1: Thank you for your valuable comments. We revised and added sentences in line 124-128. Added sentences are as following: Our results showed significant higher clinical cure rate at 10 days compared with those of previous studies, suggesting that vancomycin monotherapy would be better for the treatment of patients with severe CDI. This discrepancy could be attributed to difference in study population. However, better treatment response should be confirmed in further well designed prospective study.

Comments 2: In my opinion, the four independent factors identified as being associated with 60-day mortality and colectomy (heart rate, blood urea nitrogen level, albumin level, and hematologic malignancy) should also be mentioned in the discussion chapter.

Response 2: We appreciate the reviewers’ valuable comment. We added sentences regarding this point in line 128-134. Added sentences are as following: In addition, four factors including increased heart rate, high blood urea nitrogen level, low serum albumin level, and hematologic malignancy were independently associated with 60-day death or colectomy in multivariable analyses. These findings were similar with results from a previous study that the variables most strongly associated with death or colectomy included increased baseline comorbidities (Charlson comorbidity index >7 points), hospitalization in the ICU at the time of index C. difficile test, and low serum albumin level [9].

Reviewer 2 Report

Comments and Suggestions for Authors

This paper presents a multicenter, retrospective, observational cohort study on combination therapy for the treatment of severe CDI. The study is of interest since there are controversial results on this topic, and guidelines differ.

Some comments to the authors:

- A high percentage (almost one half) of patients received combination therapy in this multicenter study. Please provide some insight into this.

- Line 252: “patients were excluded if… received CDI treatment more than 2 days before diagnosis” Please explain

- I understand not all patients were admitted to an ICU, please confirm. 

- Information about oral tolerance is relevant in these patients, since one could think that the addition of IV agents is warranted in severe patients with no oral intake.

- Why did the authors choose to analyze in-hospital all-cause mortality and 60-day colectomy as a composite outcome?

- There was only difference in the the clinical cure rate. The definition of clinical cure seems too broad, and why did the authors choose a 10-day time?

- How did the researchers perform the telephone contact in this retrospective study for the patients discharged before the end of follow-up?

- There was a difference in hemoglobin levels between the patients who received monotherapy vs. those who received combination therapy (10.6 vs 6.3, p=0.062). The difference, although not statistically significant, seems to have clinical significance. Where the later patients with more severe conditions, which could have explained the need for IV therapy?

- Please check line 91.

- Albumin and haemathologic malignancies were relevant factors in the multivariate analysis. (Heart rate and BUN, however, showed very Little OR). Does is merit a comment in the Discusion section?

- Line 224: “Patients with severe, complicated of fulminant CDI were excluded from our study” Please check this sentence.

Author Response

Comments 1: A high percentage (almost one half) of patients received combination therapy in this multicenter study. Please provide some insight into this.

Response 1: Thank you for your valuable comments. We added sentences regarding this point in line 233-236. Added sentences are as following: Third, relatively high percentage of patients with severe CDI in our study had combination therapy. However, percentage of combination therapy was not much different between our study and previous studies comparing the effect of vancomycin monotherapy and combination therapy with IV metronidazole [8, 9, 11].

Comments 2: Line 252: “patients were excluded if… received CDI treatment more than 2 days before diagnosis” Please explain

Response 2: We excluded CDI patients with treatment including oral vancomycin more than 2 days before CDI diagnosis because larger portion of such patients had empirical treatment and stopped treatment after correct diagnosis. We would like to avoid selection bias.

Comments 3: I understand not all patients were admitted to an ICU, please confirm.

Response 3: Thank you for your valuable comments. We added data regarding ICU admission in Table 1.

Comments 4: Information about oral tolerance is relevant in these patients, since one could think that the addition of IV agents is warranted in severe patients with no oral intake.

Response 4: We appreciate the reviewers’ valuable comment. Some patients with no oral intake had oral vancomycin with L-tube. There were no patients with rectal administration of vancomycin in our cohort. We managed CDI patients with combination therapy with IV metronidazole according to physician’s discretion considering severity of CDI and patients’ condition.

Comments 5: Why did the authors choose to analyze in-hospital all-cause mortality and 60-day colectomy as a composite outcome?

Response 5: We added a sentence regarding primary outcome in line 286-287. Added sentence is as following: Composite outcome included colectomy since patients with CDI who underwent colectomy had mortality rate of 50% over [35].

Comments 6: There was only difference in the clinical cure rate. The definition of clinical cure seems too broad, and why did the authors choose a 10-day time?

Response 6: Thank you for your valuable comments. We described definition of clinical cure in detail in line 289-290. Added sentence is as following: defined as having less than 3 unformed bowel movements in 24 hours within 10 days of treatment and sustained response until at least 10 days of treatment.

Comments 7: How did the researchers perform the telephone contact in this retrospective study for the patients discharged before the end of follow-up?

Response 7: Survival status of almost patients was identified in EMR records. We contacted a few patients or relatives with telephone for identification of their survival and checking condition although design of this study is retrospective.

Comments 8: There was a difference in hemoglobin levels between the patients who received monotherapy vs. those who received combination therapy (10.6 vs 6.3, p=0.062). The difference, although not statistically significant, seems to have clinical significance. Where the later patients with more severe conditions, which could have explained the need for IV therapy?

Response 8: We appreciate the reviewers’ valuable comment. We found the mean, standard deviation, and statistical value of hemoglobin level was incorrectly calculated. We revised it in table 1. Thank you again for comment regarding this point.

Comments 9: Please check line 91.

Response 9: Sentences in line 91 was remained because we didn’t delete it in manuscript format. We deleted line 91.

Comments 10: Albumin and haemathologic malignancies were relevant factors in the multivariate analysis. (Heart rate and BUN, however, showed very Little OR). Does is merit a comment in the Discusion section?

Response 10: We appreciate the reviewers’ valuable comment. We added sentences regarding this point in line 128-134. Added sentences are as following: In addition, four factors including increased heart rate, high blood urea nitrogen level, low serum albumin level, and hematologic malignancy were independently associated with 60-day death or colectomy in multivariable analyses. These findings were similar with results from a previous study that the variables most strongly associated with death or colectomy included increased baseline comorbidities (Charlson comorbidity index >7 points), hospitalization in the ICU at the time of index C. difficile test, and low serum albumin level [9].

Comments 11: Line 224: “Patients with severe, complicated of fulminant CDI were excluded from our study” Please check this sentence.

Response 11: Thank you for your valuable comments. We corrected that sentence.